# Metatranscriptomic Analysis Reveals an Imbalance of Hepatopancreatic Flora of Chinese Mitten Crab *Eriocheir sinensis* with Hepatopancreatic Necrosis Disease

**DOI:** 10.3390/biology10060462

**Published:** 2021-05-23

**Authors:** Zeen Shen, Dhiraj Kumar, Xunmeng Liu, Bingyu Yan, Ping Fang, Yuchao Gu, Manyun Li, Meiping Xie, Rui Yuan, Yongjie Feng, Xiaolong Hu, Guangli Cao, Renyu Xue, Hui Chen, Xiaohan Liu, Chengliang Gong

**Affiliations:** 1School of Biology and Basic Medical Science, Soochow University, Suzhou 215123, China; Zeenshen0719@163.com (Z.S.); drkumarindia@163.com (D.K.); Yanbingyu0224@163.com (B.Y.); g1102851085@163.com (Y.G.); limy933@163.com (M.L.); xiemiep@163.com (M.X.); yjfeng@suda.edu.cn (Y.F.); xlhu2013@suda.edu.cn (X.H.); guanglicao@163.com (G.C.); xuery@suda.edu.cn (R.X.); 2School of Studies in Zoology, Jiwaji University, Gwalior 474011, India; 3Jiangsu Center for Control and Prevention of Aquatic Animal Infectious Disease, Nanjing 210036, China; lxmlxmeng@163.com (X.L.); jsscykzx@163.com (P.F.); yr8624@163.com (R.Y.); chenhuijsbf@163.com (H.C.); xiaohanliu1975@163.com (X.L.); 4Agricultural Biotechnology Research Institute, Agricultural Biotechnology and Ecological Research Institute, Soochow University, Suzhou 215123, China

**Keywords:** *Eriocheir sinensis*, hepatopancreas necrosis disease, metatranscriptomics sequencing, hepatopancreatic flora

## Abstract

**Simple Summary:**

The cause of Chinese mitten crab Eriocheir sinensis hepatopancreas necrosis disease (HPND) remains a mystery. In this study, metatranscriptomics sequencing was conducted to characterize the changes in the structure and gene expression of hepatopancreatic flora of crabs with and without typical symptoms of HPND; an imbalance of hepatopancreatic flora can be found in the crab with HPND, and the detected microbial taxa decreased, whereas the prevalence of Spiroplasma eriocheiris significantly increased in the hepatopancreatic flora of crabs with typical symptoms of HPND, and the relative abundances of the virus and microsporidia in crabs with HPND were very low and did not increase with disease progression. The differentially-expressed genes (DEGs) in hepatopancreatic flora between crabs with and without HPND were enriched ribosome, retinol metabolism, and biosynthesis of unsaturated fatty acid KEGG pathways. These results suggested that an imbalance of hepatopancreatic flora was associated with crab HPND, and the enriched pathways of DEGs were associated with the pathological mechanism of HPND.

**Abstract:**

Hepatopancreas necrosis disease (HPND) of the Chinese mitten crab Eriocheir sinensis causes huge economic loss in China. However, the pathogenic factors and pathogenesis are still a matter of dissension. To search for potential pathogens, the hepatopancreatic flora of diseased crabs with mild symptoms, diseased crabs with severe symptoms, and crabs without visible symptoms were investigated using metatranscriptomics sequencing. The prevalence of Absidia glauca and Candidatus Synechococcus spongiarum decreased, whereas the prevalence of Spiroplasma eriocheiris increased in the hepatopancreatic flora of crabs with HPND. Homologous sequences of 34 viral species and 4 Microsporidian species were found in the crab hepatopancreas without any significant differences between crabs with and without HPND. Moreover, DEGs in the hepatopancreatic flora between crabs with severe symptoms and without visible symptoms were enriched in the ribosome, retinol metabolism, metabolism of xenobiotics by cytochrome P450, drug metabolism—cytochrome P450, biosynthesis of unsaturated fatty acids, and other glycan degradation. Moreover, the relative abundance of functions of DEDs in the hepatopancreatic flora changed with the pathogenesis process. These results suggested that imbalance of hepatopancreatic flora was associated with crab HPND. The identified DEGs were perhaps involved in the pathological mechanism of HPND; nonetheless, HPND did not occur due to virus or microsporidia infection.

## 1. Introduction

The Chinese mitten crab (*Eriocheir sinensis*) is one of the important crustaceans with great economic value [1,2]. Crab hepatopancreatic necrosis disease (HPND) had a high mortality rate of about 40–50%, which directly affects the crab farmers [1]. The typical clinical symptoms of crabs with HPND are hepatopancreas degeneration and atrophy, muscle atrophy, and a change in the color of the hepatopancreas from golden to pale yellow and white. The gastrointestinal tract is collapsed, and clear dropsy is observed in the inner cavity of the crab as the disease progresses [2]. Diseased crabs can survive for a long time, but have no commercial value because of the low content of lipids and proteins and the low growth rate. 

Crab HPND was suggested to be caused by infection of microsporidian *Hepatospora eriocheir* [1]; however, *H. eriocheir* was not detected in all crabs with HPND [3], and the artificially-infected crabs with *H. eriocheir* had no typical HPND symptoms [4]. *Vibrio* was considered to be a causative agent of acute HPND of cultured shrimp, but significant changes in the relative abundance of hepatopancreatic flora were not found in crabs with and without HPND [5,6,7,8]. *Vibrio* was isolated from the hemolymph of crabs with HPND. However, typical symptoms of HPND were not generated in animal regression tests performed using *Vibrio*; therefore, *Vibrio* is not thought to be associated with HPND of crab [5,6,7]. Moreover, HPND of crab was not generated by injecting bacteria-free supernatants from the hepatopancreas of crabs with HPND into healthy crabs, suggesting HPND was not caused by viral infections [2,8].

A previous study found that symptoms of crab HPND could be caused by breeding crabs in water with a pH of 9.5–10.0 [2], and crabs exposed to low concentrations of insecticides caused clinical symptoms of HPND [9]. Epidemiological investigations suggest that HPND in mitten crabs may result from high a pH in surrounding waters, large aquatic plants and an abundance of cyanobacteria, or hypoxia and pesticide residues [10,11,12,13]. The association between 55 variables and HPND was assessed by a cross-sectional study method, and 11 risk factors were found to have the greatest impact on HPND prevalence, including “Recent pH in the pond”, “Frequency of the abamectin use”, “Frequency of switching aerator on in the farm”, “Frequency of disinfectant use”, “Amount of edible animal ingredient”, “Abundance of Cyanobacteria in the pond”, and “Frequency of clearing the ponds” [14].

Increasing evidence indicated that Omics provides new clues in the understanding of the etiology and pathogenesis. A metatranscriptomic survey revealed changes in the hepatopancreatic flora of the crab with HPND, but there were no statistically significant difference in viral and microsporidia communities in the hepatopancreas of crab with or without HPND [15]. Metabolomics was used to screen potential causative agents of crab HPND, fatty acid metabolic abnormalities were found in the hepatopancreas of crab with HPND, and high concentrations of propamocarb (a widely used pesticide in vegetables) were detected in the hepatopancreas of crab with HPND, suggesting that pesticide could likely be associated with HPND [12]. The transcriptomic analysis of hepatopancreatic crab with HPND and without HPND showed that the metabolism of xenobiotics by cytochrome P450, drug metabolism-cytochrome P450, chemical carcinogenesis, and material metabolism were the top five significantly enriched pathways for DEGs. The material metabolic abnormalities and drug effects from the external environment were suggested to be associated with crab HPND [16]. Our results obtained from the transcriptomic analysis indicated that crab HPND may be the result of autophagy and apoptosis, the hepatopancreas of crabs with HPND turn from golden yellow/light yellow to almost white was associated with retinol metabolism dysregulation [17].

Although the causes have been discussed from epidemiology, pathogenic microbiology and molecular pathology, the cause of crab HPND is still under debate, and pathological mechanisms of HPND are unknown. Moreover, in aquatic animal diseases, the animal with the same disease may have different symptoms, and the same symptoms may be caused by different diseases. In this study, we hypothesized that the imbalance of hepatopancreatic flora was associated with crab HPND, and the pathological changes of hepatopancreas were involved in the differentially expressed genes (DEGs) in hepatopancreatic flora between crab with and without HPND; therefore, metatranscriptomic sequencing was conducted to characterize the changes in the structure and gene expression of the hepatopancreatic flora of crabs with and without typical symptoms of HPND. Consequently, an imbalance of hepatopancreatic flora was found in the crab with HPND. The relative abundances of virus and microsporidia in crabs with HPND were very low and did not increase with disease progression. The DEGs in hepatopancreatic flora between crab with and without HPND were enriched ribosome, retinol metabolism, and biosynthesis of unsaturated fatty acid KEGG pathways. These results suggested that an imbalance of hepatopancreatic flora was associated with crab HPND and the enriched pathways were associated with the pathological mechanism of HPND.

## 2. Materials and Methods

### 2.1. Crabs 

Chinese mitten crab *Eriocheir sinensis* with HPND cannot be artificially generated in the laboratory, because the etiology and pathogenesis of HPND are unknown [2,8]. Therefore, two crabs (body weight: 100–150 g) were characterized by degeneration and atrophy of hepatopancreas, where the color of hepatopancrea changes from golden yellow to white: two crabs (body weight: 100–150 g) with mild signs of HPND, where their hepatopancreas were yellow and did not degenerate significantly, and two crabs (body weight: 100–150 g) without visible signs of HPND were collected from Anfeng town of Xinghua city, Jiangsu province, China in 2017. 

### 2.2. cDNA Library Preparation and Metatranscriptomic Sequencing

The crabs for metatranscriptomic sequencing were sampled in the same pond at the same time from Anfeng town. Total RNA was isolated from hepatopancreases (1 g) of 2 crabs without visible signs of HPND (Figure 1A) (healthy crabs), light-yellow hepatopancreases (1 g) from two crabs with mild signs (Figure 1B) (diseased crabs with mild signs) and milky white hepatopancreases (1 g) from two crabs with severe signs (Figure 1D) (diseased crabs with severe signs) using RNeasyR Plus Mini Kits (Qiagen, Valencia, CA, USA) according to the manufacturer’s protocol. After genomic DNA was removed by treatment with RNase free DNase (Qiagen, Valencia, CA, USA), the quality and quantity of total RNA were estimated with a NanoDrop 2000 Spectrophotometer (Thermo Scientific, Wilmington, USA) and an Agilent 2100 Bioanalyzer (Agilent Technologies, Palo Alto, CA, USA). RNA integrity number (RIN) was evaluated by electrophoresis on 1% agarose gel. Ribosomal RNA was washed out using the Ribo-Zero^TM^ Magnetic Kit (Epicenter, Charlotte, NC, USA). The cDNA libraries were constructed using TruSeq^TM^ RNA Sample Prep Kits (Illumina, San Diego, CA, USA). The quality of the cDNA libraries was assessed by the Agilent 2100 Bioanalyzer and the complete library was sequenced by Allwegene Technology Co., Ltd. (Nanjing, China) on a HiSeq 2500 Sequencer (Illumina, San Diego, CA, USA) using Mid Output Kits, and 150-bp paired-end reads were obtained for each run. All sequencing data were deposited in the NCBI BioSample database under accessions: SRX6579474 for healthy crabs, SRX6579475 for diseased crabs with mild signs and SRX6579476 for diseased crabs with severe signs.

### 2.3. Meta-Transcriptomic Data Analysis

#### 2.3.1. Data Preprocessing

The FastQC toolkit (http://www.bioinformatics.babraham.ac.uk/projects/fastqc/, accessed on 20 June 2019) was used to evaluate the quality of raw reads as Phred score. After eliminating adaptors with SeqPrep software (https://github.com/jstjohn/SeqPrep, accessed on 20 June 2019), low-quality bases (Phred score < 20) were trimmed and reads shorter than 50 bp were discarded using Sickle software (https://github.com/najoshi/sickle, accessed on 21 June 2019). High-quality reads were obtained by discarding rRNA reads after alignment to SILVA SSU (16S/18S) and SILVA LSU (23S/28S) databases with SortMeRNA software (http://bioinfo.lifl.fr/RNA/sortmerna/, accessed on 21 June 2019). The generated high-quality reads were then used for de novo assembly. 

#### 2.3.2. De Novo Assembly and ORFs Prediction

High-quality reads were taken in the de novo assemblies with Trinity software (http://trinityrnaseq.github.io/, version trinityrnaseq-r2013-02-25, accessed on 22 June 2019) using default parameters following the previous report [18]. Trans Gene Scan software (http://sourceforge.net/projects/transgenescan/, accessed on 22 June 2019) was applied for the ORF (open reading frame) prediction and CD-HIT software (http://www.bioinformatics.org/cd-hit/, accessed on 22 June 2019) was used to construct nonredundant gene catalogues with identity 95% and coverage 90%. 

#### 2.3.3. Gene Expression Level

Gene expression levels were assessed using fragments per kilobase of exon per million fragments mapped (FPKM) values obtained using RNA-Seq by expectation maximization (RSEM) software (http://deweylab.biostat.wisc.edu/rsem/, accessed on 5 July 2019). The empirical analysis of digital gene expression data in R (edger) software (http://www.bioconductor.org/packages/release/bioc/html/edgeR.html, accessed on 10 July 2019) was used to identify differentially expressed genes (DEGs) with false discovery rate (FDR) <0.05 and |log2FC| > 1 [19,20].

#### 2.3.4. Species Information and Taxonomic Abundance

All genes were aligned against the integrated NCBI NR database with an expectation value of 1e-5 (BLAST Version 2.2.28+, USA, http://blast.ncbi.nlm.nih.gov/Blast.cgi, accessed on 11 July 2019). Species taxonomic information was obtained from the respective taxonomy annotations in NR databases. Species abundance was assessed by calculating FPKM values for the gene in each species and taxonomic abundance was calculated at different taxonomic levels. The taxonomic abundance in each specimen was calculated at the class, order, family, genera, and species levels. Biological replicates were not set; therefore, differences in abundance profiles at each level between two groups were identified with method = “blind”, sharingMode = “fit-only” by DEGseq soft [21].

### 2.4. Functional Annotations

To understand the functions of the predicted gene, gene ontology (GO) annotations were obtained from Blast2go (https://www.blast2go.com/, accessed on 12 July 2019) with default parameters based on SWISSPROT (https://web.expasy.org/docs/swiss-prot_guideline.html, accessed on 12 July 2019) annotations. GO terms were classified using Web Gene Ontology (WEGO) annotation software (http://wego.genomics.org.cn/, accessed on 12 July 2019), GO terms with a *p*-value ≤ 0.05 were designated as a significantly enriched term for DEGs. The Kyoto Encyclopedia of Genes and Genomes (KEGG) (http://www.genome.jp/kegg/, accessed on 12 July 2019) was used for the systematic analysis of gene functions [22,23]. Pathways with a *p*-value ≤ 0.05 were designated as significantly enriched pathways for DEGs. Evolutionary genealogy of genes: Non-supervised Orthologous Groups (eggNOG) [24] (http://eggnog5.embl.de/download/eggnog_5.0/, accessed on 13 July 2019) and Carbohydrate-Active enzymes (CAZy) Database (http://www.cazy.org/, accesed on 14 July 2019) were used to annotate gene functions [25]. Cluster analysis was based on the Bray–Curtis distance to assess the similarities between samples (www.microbiomeanalyst.ca, accessed on 16 July 2019) [26].

### 2.5. PCR Detection and Sanger Sequencing

To validate the sequences determined by metatranscriptomic sequencing, several primer pairs were designed and synthesized based on the sequences obtained from metatranscriptomic sequencing (Appendix A). The extracted total RNAs (5 μg), respectively, from hepatopancreases of crabs, sampled from healthy crabs, diseased crabs with mild signs, and diseased crabs with severe signs, were used for RT-PCR. PCR products were recovered and cloned into a pMD-18-T (Takara, Dalian, China) for Sanger sequencing and obtained sequences were compared with corresponding sequences determined by metatranscriptomic sequencing.

## 3. Results

### 3.1. Sign of Crabs with HPND

Crab HPND may be explored in the terms of the health of farmed crabs, pathogens, pesticides, feed, and ecology. However, the main cause is still unknown. Crab samples with HPND were collected in the different endemic areas. The hepatopancreas of healthy crabs is plumpy and golden in color (Figure 1A). The diseased crabs showed a slow response and poor mobility. The common clinical signs are hepatopancreas degeneration and atrophy, and a change in the color of the hepatopancreas (Figure 1). In the early onset, the hepatopancreas is pale gold (Figure 1B); then hepatopancreas begins to degeneration (Figure 1C). Further, with the development of the disease course, the hepatopancreas turns white and gradually atrophies and erodes (Figure 1D,E). The hepatopancreas of crabs with severe signs of HPND eventually disappear in the later stage of the onset (Figure 1F).

### 3.2. Hepatopancreatic Flora of Crabs with HPND

To find potential pathogens of HPND crab, metatranscriptomic sequencing was conducted to assess changes in the hepatopancreatic flora of the collected crabs with HPND from Anfeng Town of Xinhua city in 2017. After removing low-quality sequencing, 32,807,952 clean reads were obtained from healthy crabs, 29,532,670 from diseased crabs with mild signs, and 27,582,606 from diseased crabs with severe signs (Appendix A). A total of 40,966 transcripts were assembled using clean data and 35,230 unigenes were obtained. Microbial taxonomic information was obtained from the taxonomy annotation NR database using alignment analysis. Species abundance was estimated based on FPKM value (Table 1). A notable change in the number of detected taxa in the hepatopancreatic flora between healthy crabs and diseased crabs with mild signs was not observed; however, taxa were increased in diseased crabs with severe signs.

The top 10 taxa in relative abundance at different taxonomic levels are in Figure 1. As a whole, the relative abundance of the bacteria belonging to *Proteobacteria* and *Mucoromycota* phyla decreased; moreover, a relative abundance of the bacteria belonging to phylum *Basidiomycota* increased with disease progression, and the relative abundance of phylum *Tenericutes* increased by 106 times in diseased crabs with severe signs (Figure 2A).

At the class level, in the healthy crabs, diseased crabs with mild signs, and diseased crabs with severe signs, the most predominant classes were *GammaP**roteobacteria* (3.0875%, 2.7123%, and 2.0248%), followed by *Mucoromycota*-Unclassified (0.8729%) and Unclassified-Unclassified (0.6064%) in healthy crabs, *Mucoromycota*-Unclassified (0.6520%) and *Eurotiomycetes* (0.2555%) in diseased crabs with mild signs, and *Mollicutes* (1.1576%) and *Unclassified-Unclassified* (0.5846%) in diseased crabs with severe signs. Compared to healthy crabs, the relative abundance of *Mollicutes* in diseased crabs with severe signs increased by 106 times (Figure 2B).

In the case of order, the abundance of *Gamma**Proteobacteria*-Unclassified was maximum (1.4327%) in healthy crabs, followed by *Pseudomonadales* (0.8613%) and *Mucorales* (0.8316%). The relative abundances of these three orders was, respectively, 1.4975%, 0.6911%, and 0.6200% in diseased crabs with mild signs, and 1.2718%, 0.3552%, and 0.2702% in diseased crabs with severe signs. The relative abundance of *Entomoplasmatales* in diseased crabs with severe signs increased 987 times compared to healthy crabs (Figure 2C).

In family, the top 3 families for relative abundance in both healthy crabs and diseased crabs with mild signs were Unclassified-Unclassified (1.4327% for healthy crabs, 1.4975% for diseased crabs with mild signs), *Pseudomonadaceae* (0.8580%, 0.6906%), and *Cunninghamellaceae* (0.7395%, 0.5681%). In diseased crabs with severe signs, the most predominant families were Unclassified-Unclassified (1.2718%), followed by *Spiroplasmataceae* (1.0452%) and *Pseudomonadaceae* (0.3521%). The relative abundance of *Spiroplasmataceae* in diseased crabs with severe signs increased by 977 times compared to healthy crabs (Figure 2D).

In the case of genus, the top 3 genera for relative abundance in both healthy crabs and diseased crabs with mild signs were Unclassified-Unclassified (1.4327% for healthy crabs, 1.4975% for diseased crabs with mild signs), *Pseudomonas* (0.8580%, 0.6906%), and *Absidia* (0.7395%, 0.5681%). In XFAF-2, the top 3 genera were Unclassified-Unclassified (1.2718%), *Spiroplasma* (1.0452%), and *Pseudomonas* (0.3521%) (Figure 2E).

At the species level, the top 3 predominant species in both healthy crabs and diseased crabs with mild signs were *Gamma Proteobacteria bacterium* 2W06 (1.4186% for healthy crabs, 1.4900% for diseased crabs with mild signs)*, Absidia glauca* (0.7395%, 0.5681%), and *Candidatus Synechococcus spongiarum* (0.2311%, 0.1457%). In XFAF-2, the top 3 predominant species were *Gamma Proteobacteria bacterium* 2W06 (1.2654%), *Spiroplasma eriocheiris* (0.5885%), and *Absidia glauca* (0.2116%) (Figure 1F). The prevalence of bacteria belonging to *A. glauca* and *C. Synechococcus spongiarum* species decreased in crabs with HPND, whereas the prevalence of *S. eriocheiris* species in diseased crabs with severe signs increased by 697 times compared to healthy crabs (Figure 2F).

At different taxonomic levels, 127 families, 141 genera, and 165 species were shared by healthy crabs, diseased crabs with mild signs and diseased crabs with severe signs. A total of 6 families in healthy crabs, 5 in diseased crabs with mild signs, and 4 in diseased crabs with severe signs; 9 genera in healthy crabs, 7 in diseased crabs with mild signs, and 4 in diseased crabs with severe signs; and 8 species in healthy crabs, 11 in diseased crabs with mild signs, and 8 in diseased crabs with severe signs (Appendix A) were specifically recorded. Although differences in the composition of hepatopancreatic flora in crabs without and with HPND were found, microbial taxa specifically detected in crabs with HPND were not dominant, their relative abundances were very low and did not increase with disease progression.

### 3.3. HPND Is Associated with a Change in the Construction of Hepatopancreatic Flora

Cluster analysis based on the top 10 taxa by relative abundance was performed using Bray–Curtis distance matrixes to assess the similarity of samples in the hepatopancreatic flora at different taxonomic levels. Diseased crabs with mild signs were similar to healthy crabs, while diseased crabs with severe signs were different from healthy crabs (data not shown). To further assess differences in hepatopancreatic flora among healthy crabs, diseased crabs with mild signs, and diseased crabs with severe signs, the top 35 genera in relative abundance were used to construct heatmaps. The hepatopancreatic flora of diseased crabs with mild signs and diseased crabs with severe signs was different from healthy crabs (Figure 3).

### 3.4. Viral Infection Is Not Involved in HPND 

The metatranscriptomic sequencing was used to find the potential pathogens of crab HPND. Homologous sequences of 27 viral genera were found in crab hepatopancreases, among the recorded genera 24 were found in healthy crabs and diseased crabs with mild signs, and 21 in diseased crabs with severe signs. The homologous sequences of specific viral genera in crabs with HPND were not found. The top 4 sequences in relative abundance were homologous to *Alphabaculovirus*, Unclassified genera, *Nepovirus,* and *Alpharetrovirus*; however, their relative abundances in diseased crabs with mild signs and diseased crabs with severe signs were lower compared to healthy crabs (Appendix A). At the species level, homologous sequences of 34 viral species were found in crab hepatopancreases. Out of 34 species, 31 were in healthy crabs, 28 in diseased crabs with mild signs, and 25 in diseased crabs with severe signs. The top 3 sequences in relative abundance were the homologous sequences of the cherry leaf roll virus, avian leukosis virus, and *Penaeus*
*monodon* nudivirus in healthy crabs; avian leukosis virus, *Heliothis virescens* ascovirus 3a, and Tanapox virus in diseased crabs with mild signs; and avian leukosis virus, *P. monodon* nudivirus, and reticuloendotheliosis virus in diseased crabs with severe signs. The relative abundances of these viral homologous sequences in crabs with HPND were low and did not increase with disease progression (Table 2), therefore crabs with HPND did not involve in viral infection.

### 3.5. Microsporidia Infection Is Not Involved in HPND 

Homologous sequences of the phylum Microsporidia that included *Mitosporidium*, *Nosema* and *Anncaliia* genera were detected in crab hepatopancreas; their relative abundance in crabs with HPND did not show any significant differences between groups (Table 3). As *H. eriocheir* was supposed to be a pathogen of crab HPND. Results of metatranscriptomic sequencing did not record *H. eriocheir*.

### 3.6. The Number of DEGs in Hepatopancreatic Flora Increased with HPND Progression

To explore the association of gene expression profiles of hepatopancreatic flora with HPND, DEGs with FDR < 0.05 and |log2FC| > 1 were identified using empirical analysis of digital gene expression data without biological replicates in edgeR software. Linking diseased crabs with mild signs and healthy crabs, data showed 489 DEGs (up: 283, down: 206); diseased crabs with severe signs and healthy crabs showed 968 (up: 310, down: 658), and diseased crabs with severe signs and diseased crabs with mild signs disclosed 673 (up: 207, down: 466) (Figure 4). Venn diagrams showed 45 DEGs were shared by diseased crabs with mild signs vs. healthy crabs, diseased crabs with severe signs vs. healthy crabs, and diseased crabs with severe signs vs. diseased crabs with mild signs. In common, were 6 upregulated genes and 12 downregulated genes (Figure 5). DEGs heat maps showed differences in gene expression profiles in hepatopancreatic flora among healthy crabs, diseased crabs with mild signs and diseased crabs with severe signs. Among all, healthy crabs and diseased crabs with mild signs were grouped, and diseased crabs with severe signs were in another group, and the number and expression level of DEGs changed as disease progressed (Appendix A).

### 3.7. The Function of Differentially Expressed Genes (DEGs) Associated with HPND Pathological Mechanism 

To understand the pathological mechanism of HPND, GO enrichment of DEGs was performed. DEGs for diseased crabs with mild signs vs. healthy crabs were enriched to 30 GO terms such as metabolic, organic substance metabolic, and primary metabolic processes in the biological process (BP); intracellular, organelle, and intracellular organelle in the cellular component (CC); and structural molecule activity and structural constituent of the ribosome in molecular function (MF) (Figure 6A). DEGs for diseased crabs with severe signs vs. healthy crabs were enriched to 28 GO terms: metabolic process, biosynthetic process, organic substance biosynthetic process in BP; cytoplasm, the cytoplasmic part in CC; and oxidoreductase activity and structural molecule activity in MF (Figure 6B); DEGs for diseased crabs with severe signs vs. diseased crabs with mild signs were enriched to 17 GO terms: metabolic process, single-organism metabolic process and oxidation-reduction process in BP, and catalytic activity, oxidoreductase activity, and hydrolase activity in MF (Figure 6C). Three enriched GO terms (hydrolase activity-hydrolyzing O-glycosyl compounds, carbohydrate metabolic process, and hydrolase activity-acting on glycosyl bonds) were shared by diseased crabs with mild signs and healthy crabs, diseased crabs with severe signs and healthy crabs, and diseased crabs with severe signs and diseased crabs with mild signs. These results indicated that carbohydrates in the hepatopancreas of crab were used by hepatopancreatic flora.

KEGG enrichment was used to annotate DEG functions. The top 20 enriched KEGG pathways are in Appendix A: ribosome and another glycan degradation for diseased crabs with mild signs vs. healthy crabs (Appendix A); ribosome, retinol metabolism, metabolism of xenobiotics by cytochrome P450, drug metabolism—cytochrome P450, and other glycan degradation for diseased crabs with severe signs vs. healthy crabs (Appendix A); and lysosome, and sphingolipid metabolism for diseased crabs with severe signs vs. healthy crabs (Appendix A). Seven enriched KEGG pathways (retinol metabolism, lysosome, other glycan degradation, metabolism of xenobiotics by cytochrome P450, drug metabolism cytochrome P450, ribosome, and steroid hormone biosynthesis) were shared by diseased crabs with mild signs and healthy crabs, and diseased crabs with severe signs and healthy crabs. Of these, the expression of all 68 enriched genes to ribosomes in diseased crabs with mild signs vs. healthy crabs and 71 in diseased crabs with severe signs vs. healthy crabs was upregulated (Appendix A), indicating that nutrients in hepatopancreas were used to synthesize microbial proteins. The expression of all 14 enriched genes to retinol metabolism in diseased crabs with severe signs vs. healthy crabs was downregulated (Figure 7), implying that the levels of retinal, retinoate, rhodopsin, and beta-carotene, which were associated with the color of hepatopancreas, decreased in the diseased crabs with severe signs. Moreover, the expression of all 7 enriched genes to the biosynthesis of unsaturated fatty acids in diseased crabs with severe signs vs. healthy crabs was downregulated (Appendix A). 

### 3.8. Relative Abundances of Functions Changed with the Pathogenesis 

Changes in functional abundance of genes in the hepatopancreatic flora were indicated in Figure 8. Compared to healthy crabs, the relative abundance of genetic information processing increased by 1.3798 times for diseased crabs with mild signs and 1.5341 times for diseased crabs with severe signs. The relative abundance of other functions except for human diseases decreased with disease progression in KEGG annotations (Figure 8A). For eggNOG, the relative abundance of translation-ribosomal structure and biogenesis increased. The relative abundance of general function prediction only and posttranslational modification-protein turnover-chaperones decreased with pathogenesis (Figure 8B). In the CAZy, the relative abundance of xenobiotics and carbohydrate esterases increased, and the relative abundance of glycoside hydrolases decreased in diseased crabs with mild signs and diseased crabs with severe signs compared to healthy crabs (Figure 8C). Clustering analysis using relative abundances of functions showed that diseased crabs with mild signs and diseased crabs with severe signs were together and the relative abundances of functions changed with pathogenesis (Appendix A).

### 3.9. PCR Detection and Sanger Sequencing

To validate the sequences detected by metatranscriptomic analysis, the homologous sequences of genes from *C. sesamiae* bracovirus, *P. monodon* nudivirus, *E. sinensis* reovirus, *M. depressus* WSSV-like virus, and *S. eriocheiris,* respectively, were amplified from the cDNA of diseased crabs with severe signs by PCR. The identified sequences of PCR products were identical with the corresponding sequences determined by metatranscriptomic sequencing. Moreover, *P. monodon* nudivirus, *E. sinensis* reovirus, and the *M. depressus* WSSV-like virus were only found in healthy crabs and diseased crabs with severe signs by PCR processes. *C. sesamiae* bracovirus and *S. eriocheiris* were found in all three samples as detected by metatranscriptomic sequencing. 

## 4. Discussion

The aetiology of HPND was explored in the pathogens [1,4], pathological changes [2,8,17], ecotype [9,10,11,14], hepatopancreatic flora [15], the transcriptome of hepatopancreas [1,6,17], metabolite profiling of the hepatopancreas [12], and offspring seed and epidemiological surveys [8,13]; however, the aetiology of HPND crab remains unidentified.

The etiology of aquatic animal diseases can be roughly divided into biological and abiotic factors. *H. eriocheir* belonging to microsporidian was considered as the pathogen of crab HPND, because *H. eriocheir* was found in the crabs with HPND [1]. However, *H. eriocheir* was not found in some crabs with HPND [2,17,27]. The typical HPND symptoms could not be generated by artificial infection with *H. eriocheir* [4], Moreover, the sequences of *H. eriocheir* were not found in the hepatopancreatic flora by a meta-transcriptomic survey [15]. In the present study, the *Mitosporidium*, *Nosema,* and *Anncaliia* genera belonging to phylum *Microsporidia* were recorded in hepatopancreatic flora of the crabs with HPND by the meta-transcriptomics data. The significant differences in relative abundances between crabs with and without HPND were not observed. Moreover, sequences of *H. eriocheir* were not identified by metatranscriptomic sequencing. Therefore, it can be concluded that crab HPND was not caused by the microsporidian infection.

To date, studies on *E. sinensis* disease caused by viral infection is limited. White spot syndrome virus [28], Reovirus [29,30,31,32,33], and Roni-like virus [34] were noticed in the diseased *E. sinensis*. However, signs of HPND were not found in the infected crabs with these viruses. Eleven families of viruses were identified in the hepatopancreata of crab by meta-transcriptome analysis; however, the expression levels of these viruses were extremely low in the crabs with and without HPND [14]. In our previous study, virus-like particles were not found in the hepatopancreatic cells of crabs with HPND by electron microscope observation [2,17]. Moreover, HPND was not generated by inoculation with bacteria-free supernatants from the hepatopancreas of the crabs with HPND [2]. In this study, homologous sequences of 34 viral species were identified in the hepatopancreases of crab from the analysis of the meta-transcriptome data; however, the relative abundances of these viral homologous sequences in crabs with HPND were very low and did not increase with disease progression. These results suggested that crab HPND was not caused by viral infection. However, the identified sequences by metatranscriptomic sequencing are derived from transcripts of microbial genes, suggesting that the detected viruses, especially DNA viruses, infect crab. Our findings provide clues for molecular epidemiological investigations of crab diseases.

The *Vibrio* is considered as the causal agent of acute HPND of cultured shrimp [5,6,7,8]. Bacteria belonging to *Vibrio* spp. could be 10~23.33% in crabs with HPND; however, HPND-like signs were not generated in animal regression tests performed with *Vibrio* spp. [13]. In our findings, hepatopancreatic flora in crabs with HPND was not dominated by the genera Vibrio. Therefore, *Vibrio* spp. was not a pathogen of HPND.

Microsymbionts in the hepatopancreas of isopods were involved in digestion, nutrition and absorption, reproduction, and immunity [35,36]. The imbalance of symbionts in the hepatopancreas is associated with some illnesses [37,38,39]. Symbiotic bacteria in crabs belong to the phyla Bacteroidetes, Proteobacteria, Firmicutes, and Tenericutes [40,41,42]. The genus *Candidatus hepatoplasma* in phylum Tenericutes is beneficial to its isopod host under low-nutrient conditions [43]. The genus *Spiroplasma* belonging to phylum Tenericutes is deemed a pathogen of Chinese mitten crab tremor disease [39,44], and *Acholeplasma* spp. are linked to clearwater disease of the mud crab *Scylla serrata* [38]. A previous report indicated that the relative abundance of phylum Tenericutes increases in crabs with HPND [15]. Similar results were found in our study. The largest increase in relative abundances was *S. eriocheiris* belonging to Tenericutes. However, typical tremor symptoms were not observed in crabs with HPND. The possibility that an increase in the abundance of *S. eriocheiris* is associated with HPND should be further studied.

Some diseases are not caused by a specific microbial infection, but by an imbalance in microbial flora. Host disease can be induced by symbiotic microbes after this balance is disrupted [43,45]. Significant differences in the hepatopancreatic flora were observed between crabs with and without HPND. The microbial diversity reduced and the microbial amount increased in the hepatopancreatic flora of crabs with HPND. Specifically, the prevalence of the Tenericutes phylum increase, the prevalence of the proteobacteria and Bacteroidetes phyla decreased in crabs with HPND [15]. Similar results were found in this study. Therefore, we strongly suggested imbalance in the hepatopancreatic flora is associated with HPND. However, determining if a change in hepatopancreatic flora is the cause or result of HPND is difficult. Nonetheless, a large number of nutrients are consumed by bacteria, which may result in hepatopancreas atrophy. Apoptosis and autophagy were found in the hepatopancreases of crabs with HPND [2,17]. It has been known that both apoptosis and autophagy can be induced by nutrition deficiency [46]. Therefore, apoptosis and autophagy of hepatopancreases of crabs with HPND may associate with nutrient deficiency caused by the multiplication of symbiotic bacteria.

To explore the association of gene expression profiles of hepatopancreatic flora with HPND, DEGs in hepatopancreatic flora between crabs with and without HPND were identified by analyzing meta-transcriptomic sequencing data. The results indicated that the number of DEGs increased with HPND progression. Therefore, these DEGs could be associated with the pathological mechanism of HPND. The expression of all 71 genes enriched in ribosomes was upregulated in diseased crabs with severe signs vs. healthy crabs, which revealed that microbial protein synthesis was enhanced and host protein synthesis was obstructed in the crab hepatopancreas. This may be one of the causes of hepatopancreas degeneration of crabs with HPND. The hepatopancreas color is believed to be associated with pigments including β-carotene [17]. β-carotene, a red-orange pigment, can be transformed into retinol and retinene [47]. In this study, we found that the expression of all 14 enriched genes to retinol metabolism in diseased crabs with severe signs vs. healthy crabs was down-regulated. The color of the hepatopancreases of crabs with HPND turns from golden/light yellow to almost white during pathogenesis [1]. Therefore, we suggested that the color change of the hepatopancreases of crabs with HPND was associated with the down-regulation of genes related to retinol metabolism in the hepatopancreatic flora. Downregulated genes in the hepatopancreatic flora of crab with HPND were enriched to the biosynthesis of unsaturated fatty acids, suggesting that liposoluble carotenoid accumulation was reduced.

Crab HPND was suggested to be the result of both autophagy and apoptosis induced by some unknown abiotic factors including pesticides, toxins from environmental bacteria and algae, chemical fertilizer, antibiotics in food, and farming water [2,17]. This speculation is supported by epidemiological investigations that crab HPND may result from hypoxia and pesticide residues, high pH in surrounding waters, massive aquatic plants and abundance of cyanobacteria, or in water environments with empty-illumination, without aquatic plants and reduced feeding, or hypoxia and pesticide residues [2,4,9,11,13]. The association between 55 variables and HPND was assessed. Recent pH in the pond, frequency of the abamectin and disinfectant used, frequency of switching aerator on the farm, and abundance of Cyanobacteria in the pond etc. were found to have the greatest impact on HPND prevalence. Bacterial symbionts change with the ecological environment in which their hosts live [35,36]. Therefore, the relationships among environment, hepatopancreatic flora and crab HPND need to be explored.

Several studies have indicated that increased occurrence rates of HPND with increased pesticide applications [9,10,12]. A high concentration of propamocarb was detected in the hepatopancreas of crabs with HPND [11]. Moreover, crabs exposed to a low concentration of insecticides caused clinical symptoms of HPND [9]. Interestingly, DEGs in the hepatopancreas between crab with HPND and without HPND were enriched to the metabolism of xenobiotics by cytochrome P450, drug metabolism-cytochrome P450, and material metabolism [16]. Our previous transcriptomic analysis indicated that the down-regulated DEGs in the hepatopancreas of crabs with HPND were enriched in chlorocyclohexane and chlorobenzene degradation KEGG pathways [16]. In this study, we found that DEGs in the hepatopancreatic flora between crab with HPND and without HPND were also enriched in the metabolism of xenobiotics by cytochrome P450 and drug metabolism-cytochrome P450 KEGG pathways. A comparison of the relative abundance of functions for DEGs in the hepatopancreatic flora among samples indicated an increase in the relative abundance of xenobiotics in the diseased crabs. These results strongly implied that unknown toxic substances including pesticide residues in the aquaculture environment are associated with HPND.

## 5. Conclusions

The imbalance of hepatopancreatic flora is associated with crab HPND. However, HPND does not correlate with virus or microsporidia infection. Changes in hepatopancreatic flora and the enriched pathways of DEGs may associate with the pathological mechanism of HPND.

## Figures and Tables

**Figure 1 biology-10-00462-f001:**
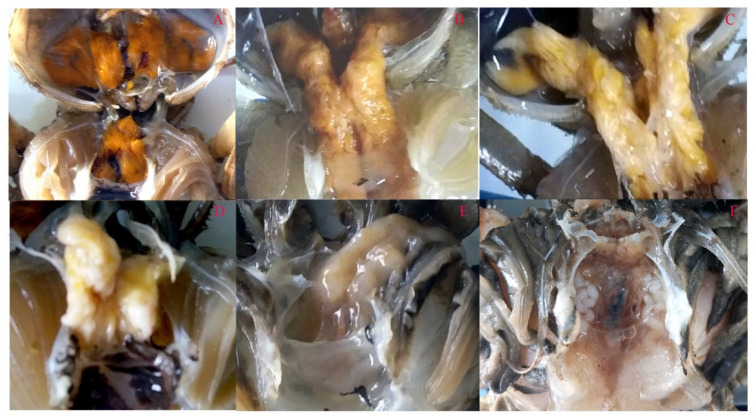
The common clinical signs of crabs with HPND. The body weights were about 100–150 g. (**A**) hepatopancreas of healthy crab, the hepatopancreas golden and plump; (**B**) hepatopancreas of crab with mild signs, the hepatopancreas are yellow and do not degenerate significantly; (**C**) the hepatopancreas turms pale yellow and begins to degeneration; (**D**) the hepatopancreas turns white and gradually atrophies; (**E**) the hepatopancreas erodes; (**F**) the hepatopancreas disappears.

**Figure 2 biology-10-00462-f002:**
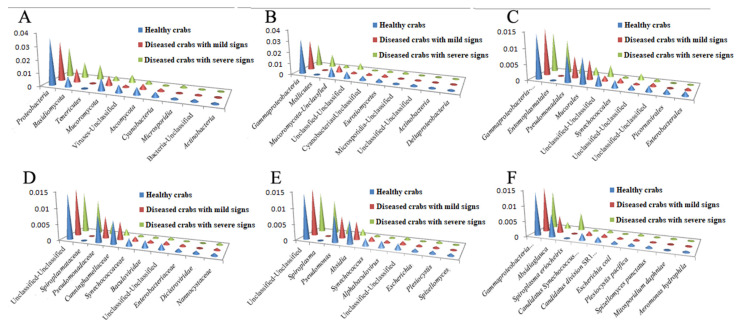
Top 10 taxon in relative abundance at different taxonomic levels. (**A**), (**B**), (**C**), (**D**), (**E**), and (**F**) represented Phyla, Classes, Orders, Families, Genera, and Species, respectively.

**Figure 3 biology-10-00462-f003:**
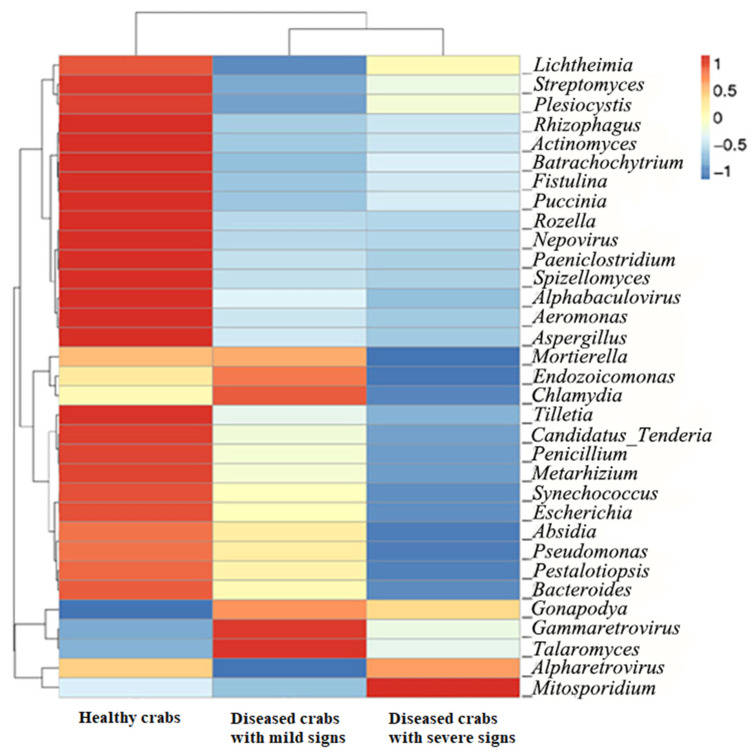
Clustering of relative abundance at the genus level. The horizontal axis represented sample information; vertical axis represented species information; the cluster tree on the left side was taxon clustering tree; the color intensity in the square grid represented the bacterial relative abundance, which is named as z-value and generated by the relative abundance of the genus in each line after normalization treatment.

**Figure 4 biology-10-00462-f004:**
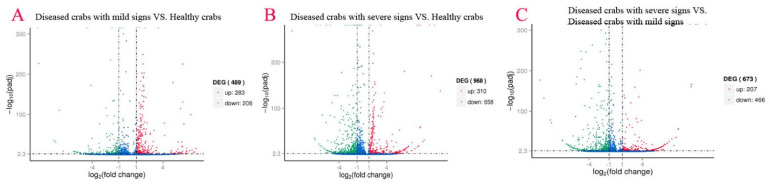
Volcano plot for gene expression among different samples. (**A**) Diseased crabs with mild signs VS. Healthy crabs; (**B**) Diseased crabs with severe signs VS. Healthy crabs; (**C**) Diseased crabs with severe signs VS. Diseased crabs with mild signs.

**Figure 5 biology-10-00462-f005:**
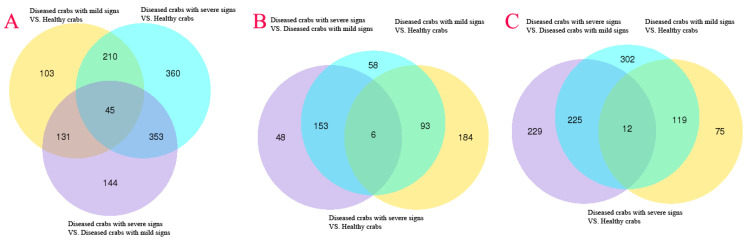
Venn diagram of DEGs identified by metatranscriptomic sequencing. (**A**) Venn diagrams of all DEGs among Diseased crabs with mild signs and Healthy crabs, Diseased crabs with severe signs and Healthy crabs, and Diseased crabs with severe signs and Diseased crabs with mild signs; (**B**) Venn diagrams of up-regulated DEGs among Diseased crabs with mild signs and Healthy crabs, Diseased crabs with severe signs and Healthy crabs, and Diseased crabs with severe signs and Diseased crabs with mild signs; (**C**) Venn diagram of down-regulated DEGs among Diseased crabs with mild signs and Healthy crabs, Diseased crabs with severe signs and Healthy crabs, and Diseased crabs with severe signs and Diseased crabs with mild signs.

**Figure 6 biology-10-00462-f006:**
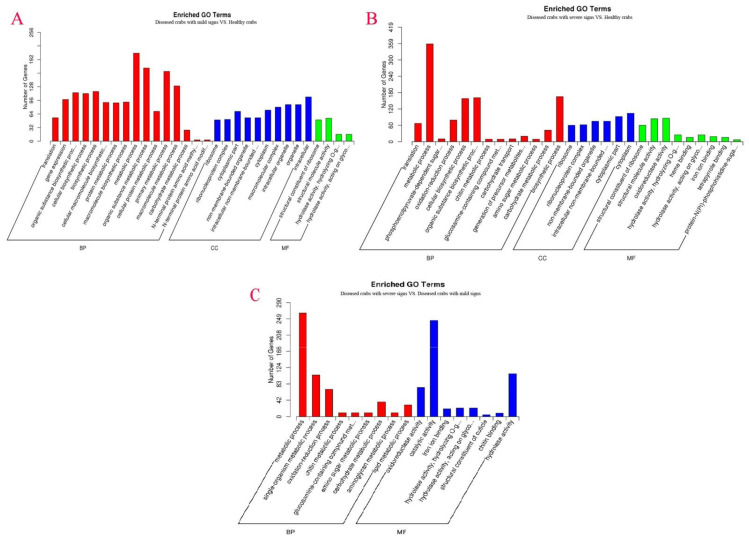
GO enrichment analysis of DEGs identified in the hepatopancreatic flora. (**A**) diseased crabs with mild signs vs. healthy crabs; (**B**) diseased crabs with severe signs vs. healthy crabs; (**C**) diseased crabs with severe signs vs. diseased crabs with mild signs. BP, CC, and MF represented biological processes, cellular components and molecular functions, respectively.

**Figure 7 biology-10-00462-f007:**
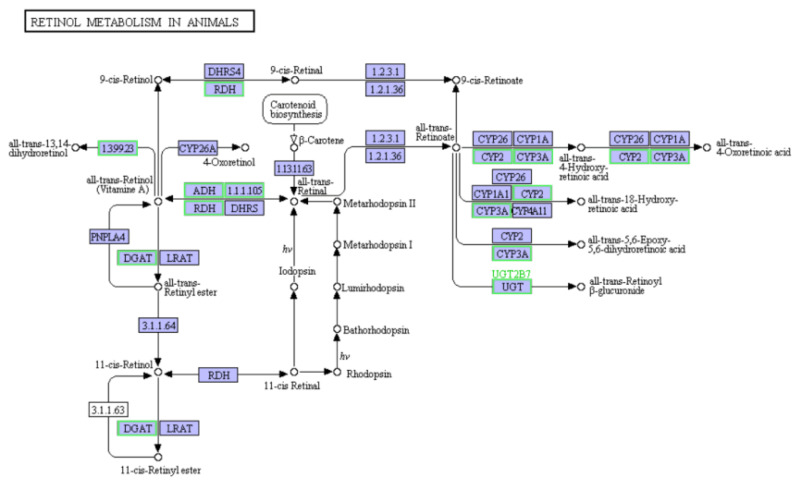
14 enriched genes to retinol metabolism in diseased crabs with severe signs vs. healthy crabs were downregulated. The green boxes represent the downregulated genes.

**Figure 8 biology-10-00462-f008:**
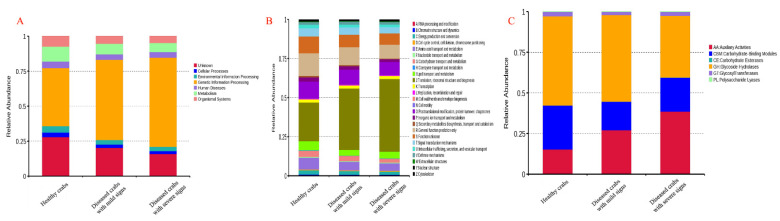
Functional abundances of different samples at eggNOG, KEGG, and CaZy levels. (**A**) KEGG; (**B**) eggNOG; (**C**) CaZy.

**Table 1 biology-10-00462-t001:** Changes in the relative abundance of microbes at different taxonomic levels in the hepatopancreatic flora of crabs with HPND.

Samples	Kingdoms (Number/Percentages)	Phyla	Classes	Orders	Families	Genera	Species
Healthy crabs	4/30.37%	33/4.48%	64/3.69%	110/3.03%	172/2.87%	197/2.78%	205/2.12%
Diseased crabs with mild signs	4/35.43%	33/3.78%	65/2.92%	110/2.47%	169/2.40%	194/2.37%	205/1.85%
Diseased crabs with severe signs	4/30.86%	33/3.41%	61/2.64%	103/2.19%	162/1.98%	180/1.96%	189/1.50%

The values in the bracket represented the percentage of the microbe which was assigned to a taxon.

**Table 2 biology-10-00462-t002:** Relative abundances of the detected viruses in the samples.

Homologous Sequences of Detected Virus	Relative Abundance (%)			Fold Change	
	Healthy Crabs	Diseased Crabs with Mild Signs	Diseased Crabs with Severe Signs	Diseased Crabs with Mild Signs/Healthy Crabs	Diseased Crabs with Severe Signs/Healthy Crabs
Cherry leaf roll virus	0.0246	0.0006	0.0002	0.0243	0.0081
Avian leukosis virus	0.0163	0.0134	0.0156	0.8221	0.9571
Penaeus monodon nudivirus	0.0161	0	0.0114	0	0.7081
Heliothis virescens ascovirus 3a	0.0065	0.0038	0.0025	0.5846	0.3846
Tanapox virus	0.0048	0.0033	0.002	0.6875	0.4167
Swinepox virus	0.0034	0.0014	0.0013	0.4117	0.3823
Reticuloendotheliosis virus	0.0026	0.0021	0.0033	0.8077	1.2692
Tipula oleracea nudivirus	0.002	0.0026	0.0017	1.3	0.85
Cotesia sesamiae bracovirus	0.0015	0.0006	0.0004	0.4	0.2667
Chelonus inanitus bracovirus	0.0008	0.0007	0.0005	0.875	0.625
Lymphocystis disease virus Sa	0.0008	0.0001	0.0001	0.125	0.125
Metopaulias depressus WSSV-like virus	0.0008	0	0.0003	0	0.375
Pigeonpox virus	0.0007	0.0002	0.0005	0.2857	0.7143
Marine RNA virus SF-1	0.0007	0.0002	0	0.2857	0
Cyprinid herpesvirus 3	0.0005	0.0003	0.0004	0.6	0.8
Saimiriine herpesvirus 4	0.0005	0.0002	0	0.4	0
Oryctes rhinoceros nudivirus	0.0004	0.0001	0.0004	0.25	1
Avian musculoaponeurotic fibrosarcoma virus AS42	0.0004	0.0001	0.0003	0.25	0.75
Murine leukemia virus	0.0003	0.0004	0.0005	1.3333	1.6667
Abelson murine leukemia virus	0.0003	0.0004	0.0004	1.3333	1.3333
Deerpox virus W-848-83	0.0003	0.0002	0	0.6667	0
Glypta fumiferanae ichnovirus	0.0003	0.0001	0.0001	0.3333	0.3333
Antarctic picorna-like virus 1	0.0003	0.0001	0.0001	0.3333	0.3333
*Eriocheir sinensis* reovirus	0.0003	0	0.0001	0	0.3333
UR2 sarcoma virus	0.0003	0	0	0	0
Aureococcus anophagefferens virus	0.0002	0.0001	0.0001	0.5	0.5
Cotesia congregata bracovirus	0.0002	0.0001	0	0.5	0
Bovine papular stomatitis virus	0.0001	0.0002	0.0008	2	8
Infectious spleen and kidney necrosis virus	0.0001	0	0.0002	0	2
Hepelivirus	0	0.0011	0	0	0
Canarypox virus	0	0.0004	0	0	0
Yaba monkey tumor virus	0	0	0.0002	0	0
Avian sarcoma virus	NA				
Alphapapillomavirus 7	NA				

NA, homologous sequences could be detected, but the abundance was very low. Relative abundance of viruses in healthy crabs was used as the control in the calculation of fold change.

**Table 3 biology-10-00462-t003:** Relative abundances of the detected microsporidium in the samples.

Detected Homologous Sequences of Microsporidian	Relative Abundance (%)		
	Healthy Crabs	Diseased Crabs with Mild Signs	Diseased Crabs with Severe Signs
Mitosporidium daphniae	8.39 × 10^−5^	3.91 × 10^−5^	2.79 × 10^−4^
Nosema bombycis	8.07 × 10^−6^	1.08 × 10^−5^	4.73 × 10^−6^
Nosema apis	7.99 × 10^−6^	6.73 × 10^−6^	1.44 × 10^−6^
Anncaliia algerae	2.13 × 10^−6^	1.47 × 10^−6^	1.64 × 10^−6^

## Data Availability

Not applicable.

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
