# Peer review of "Metatranscriptomic Analysis Reveals an Imbalance of Hepatopancreatic Flora of Chinese Mitten Crab *Eriocheir sinensis* with Hepatopancreatic Necrosis Disease"

_biology, 2021, doi:10.3390/biology10060462_

Round 1
Reviewer 1 Report
I think the authors have made some revisions that improve the manuscript. There remains some grammar issues although not many. These will be probably be corrected in preparation of proofs.
Author Response
Reviewer 1
Comments and Suggestions for Authors
I think the authors have made some revisions that improve the manuscript. There remains some grammar issues although not many. These will be probably be corrected in preparation of proofs.
Response: Thank you for your positive comments. We will check the manuscript carefully according to your requests.

Reviewer 2 Report
I have reviewed the revised manuscript by Shen et al. investigating Hepatopancreatic Necrosis Disease of Chinese Mitten Crabs. The authors have removed the ultrastructural study and concentrated on the metatranscriptome analysis. The study provides some interesting data which will be useful for further studies.
Line 60 – crab
Line 62 remove be, “typical symptoms of HPND were not generated”
Line 203 – not clear what is meant by offspring quality, health of farmed crabs?
Line 348 – suggest adding figures S3 and S4 in main paper rather than supplemental
Line 475 - comma needed after references
Author Response
Reviewer 2
Comments and Suggestions for Authors
I have reviewed the revised manuscript by Shen et al. investigating Hepatopancreatic Necrosis Disease of Chinese Mitten Crabs. The authors have removed the ultrastructural study and concentrated on the metatranscriptome analysis. The study provides some interesting data which will be useful for further studies.
Response: Thank you for your positive comments.
Line 60 – crab
Response: It has been modified.
Line 62 remove be, “typical symptoms of HPND were not generated”
Response: The ‘be’ has been removed as your suggestion.
Line 203 – not clear what is meant by offspring quality, health of farmed crabs?
Response: We didn’t have a clear description here. ‘offspring quality’ has been changed to ‘health of farmed crabs’ in the revised manuscript.
Line 348 – suggest adding figures S3 and S4 in main paper rather than supplemental
Response: We have added Figures S3 and S4 in main paper as your suggestion.
Line 475 - comma needed after references
Response: we have modified as your suggestion.

Reviewer 3 Report
Thank you for inviting me to check the revised manuscript
I read the reviewer comments and author rebuttal letter, it look good to me.
I also read the revised manuscript quickly, the suggestions were thoroughly incorporated in the manuscript.
Only two comments
Line 55 - why the surviving crabs has no commercial value, whether it is related with protein and lipid content or growth rate. Explain it.
In reference number 5- there are two references mixed together.
In my opinion, the revised manuscript can be accepted for publication.
Author Response
Reviewer 3
Comments and Suggestions for Authors
Thank you for inviting me to check the revised manuscript. I read the reviewer comments and author rebuttal letter, it look good to me. I also read the revised manuscript quickly, the suggestions were thoroughly incorporated in the manuscript.
Response: Thank you for your positive comments.
Only two comments
Line 55 - why the surviving crabs has no commercial value, whether it is related with protein and lipid content or growth rate. Explain it.
Response: Due to the low growth rate and the low content of lipid and protein in hepatopancreas of the diseased crab,the surviving crabs have no commercial value. We have explained it in the revised manuscript.
In reference number 5- there are two references mixed together.
Response: It has been modified.
In my opinion, the revised manuscript can be accepted for publication.
Response: Thank you for your positive comments

This manuscript is a resubmission of an earlier submission. The following is a list of the peer review reports and author responses from that submission.
Round 1
Reviewer 1 Report
Overall comment
The manuscript entitled “Hepatopancreatic Necrosis Disease of Chinese Mitten Crab Eriocheir sinensis not caused by Virus or Microsporidia Infection” which you submitted to 'biology', has been reviewed. I am not an expert on aetiology study field, but as a reader, I have found this manuscript interesting. However, the paper makes many statements that seem unnecessary and applies several analytical tools, but it is not clear why. The objectives and hypothesis of the study are not clearly stated, which makes the discussion difficult to follow. While I am generally positive about the manuscript, I think there are some major points that should be issued (or at least clarified) by the authors; as well as some minor comments which I hope will help improving the manuscript.
Major comments
- Author insisted on the hepatopancreatic flora caused by HPND was not due to virus or microsporidia infection by a transmission electron microscope. However, in my opinion, it is the mistake of hasty generalization. How many sample did you analyze? How many study sites did you survey? Author should change the focus of the study!
- Introduction does not include sufficiently a validity of the study. You should state objective and hypothesis of the study more clearly and should provide relevant references.
- Readers of ‘biology’ might be interesting about your methods part in details to apply your methodological approach as you mentioned if they accept merit of your study. However, your methods part is too simple and do not provide detailed information. It is important to avoid that in order to enforce the credibility of a paper that has important data that might show your objective. Therefore, you should explain it much more clearly and provide more detailed information (i.e. how many sample did you use, how much volume did you use....).
- Results part is too long and broad. You should focus on results related your objectives. Research paper is not just report some results, should focus on the objective of study! There is no scale bar in your photos (Figure 1 ~ 3). Please provide scare information…
- There must be a better way to structure your discussion. To my mind, many of the information in it should be somehow incorporated already in the introduction. This would allow the reader to understand the issues of the study. Think carefully what and when you want to say, which should improve the flow of thoughts. Now it is chaotic, information jumps right at you from places where one does not expect to find them. It also lacks wider context and including references to existing literature.
Minor comments
Please check the journal format again.
- Line 41: Eriocheir sinensis ==> Eriocheir sinensis . Please check the scientific name throughout the manuscript. It should be use an italic type.
- Line 41-42: please provide the relevant supporting references.
- Line 43-46: please provide the relevant supporting references.
- Line 72: How many sample number?
- Line 72-72: please provide the relevant supporting references.
- Line 78-80: please provide more details about electron microscopy observation.
- Line 150-152: please move it to discussion part. You should provide results of your experiment in result part.
- Line 154-155: How do you know? Please provide quantitative information of mobility such as speed or index.
- Line 190: HNPD ==> HPND??
- Figure 1: please add a scale bar.!
- Figure 2: please add a scale bar.!
- Figure 3: please add a scale bar.!
- Table 1: Do we need 4 decimal place?
- Figure 4: Which one is species level? There is no information of each panels (A, B, C…) about taxonomic levels. Please provide details in figure legend.
- Figure 5.: Some of taxonomic information does not match with species level information. For example Actinomyces sp. VUL4_3, DSM 15324, and Bacteroides_sp.SM23_62_1, these matched databases do not provide species level information. It should be interpreted as genus level information!
- Figure 6.: What are the meaning of BP, CC, MF?? Please provide this abbreviation information in the figure legend.
- Line 452: Vibrio sp ==> Vibrio spp.??
- Line 452: 10%-23.33% ==> 10~23.33%
- Line 463-464: please provide p-value to show the correlation with HPND.
- Line 470: evolution,t ==> evolution
Reviewer 2 Report
I have reviewed the manuscript by Shen et al. investigating Hepatopancreatic Necrosis Disease of Chinese Mitten Crabs. The conclusions of this paper are very similar to the work of Pan et al (2017) and Shen et al (2017), both publications reaching the same conclusions. Although presenting some interesting data I do not think the manuscript is suitable for publication in its current format and recommend that it is rejected.
Recommend manuscript is edited by native English speaker
Line 24 – This needs to be reworded, suggest Ultrastractural analysis of diseased crab tissues identified structural and …
Line 41 – background on disease is required, what are the clinical signs?
Line 47 what is hydrops?
Line 52 speciation? Can the authors clarify what they mean here
Line 63 Penaeus
Line 73 – what are the signs of infection? How was this confirmed to split the crabs between infected and uninfected? More details required
Line 77 – More detail is needed, fixative used? Processing details? Stain? Suggest authors state samples were processed as reported by Yan et al 2020, if they describe full methods. Otherwise full details required here.
Line 82 – were the same crabs sampled for TEM and sequencing?
Line 152 – germplasm? Can the authors clarify what they mean here
Line 175 – Histology is usually wax sections observed under light microscope, study here describes TEM study suggest rewording unless histology was also used? In which case please detail results
Line 176 onwards – The problem with studying tissues for presence/absence of a pathogen with TEM is the size of sample analysed, TEM takes very small pieces of tissue. We know some pathogens can be focal within tissues so could be missed when such small pieces of tissue are examined. This analysis can not be used to confirm that the tissues were pathogen free.
I also disagree with some of the conclusions and the images are not representative to display what the authors are suggesting, you ideally need lower power images to show damage, fixation of tissues looks poor and it is not clear whether it is the condition or fixation problems. Normal tissue structure needed as comparison.
In my opinion the TEM study presented here adds little to the paper and should be removed.
Tables– suggest labelling samples as healthy and diseased to assist the reader
Figure 4 and 5 – suggest labelling as above
Line 282 onwards – Interesting results, although the authors state viruses are not involved in HPND infection do the authors think the crabs are infected with these (or similar) virus infections? If so, could the presence of these viral sequences be related to possible stress and immune suppression? Are you more likely to see virus sequences in infected animals?
Line 306 – sequencing did not record H. Eriocheir but could crabs be infected with another species of microsporidian? Figure 3B could be early development stages of a microsporidian infection.
Line 385 – Do the authors think the crabs are infected with these (or similar) viruses? What is the significance of this finding?
Line 407 – as above you cannot state this based on the limited TEM study performed
Author Response
Comments and Suggestions for Authors
I have reviewed the manuscript by Shen et al. investigating Hepatopancreatic Necrosis Disease of Chinese Mitten Crabs. The conclusions of this paper are very similar to the work of Pan et al (2017) and Shen et al (2017), both publications reaching the same conclusions. Although presenting some interesting data I do not think the manuscript is suitable for publication in its current format and recommend that it is rejected.
Response: Thank you for your rigorous comments. The works reported by Pan et al.(2017) was our previous study. Ultrathin sections of different tissues from the sampled crabs with HPND from the Yangcheng lake of Suzhou city were observed with transmission electron microscope. In this study, the sampled crabs with HPND from 3 epidemic areas were inspected by a transmission electron microscope. The conclusions of the two studies are similar as you pointed out. Therefore, ultrastractural analyses of diseased crab tissues were removed in the revised manuscript according to the suggestions of two reviewers. Shen et al. reported that changes in the hepatopancreatic flora of the sampled crab with HPND from Yandu District, Yancheng City, Jiangsu Province (2017). However, function analysis (GO, KEGG, eggNOG and CAZy ) of differentially expressed genes (DEGs) in the hepatopancreatic flora between crabs with and without HPND were not conducted to explore the pathogenic mechanism of HPND in their study. In our study, the hepatopancreatic flora of diseased crabs with mild symptoms, diseased crabs with severe symptoms and crabs without visible symptoms were investigated using metatranscriptomic sequencing, not only the comparison of hepatopancreatic flora between crabs with and without HPND at different taxonomic levels was performed, but also function analysis of DEGs in the hepatopancreatic flora between crabs with and without HPND. Our results provided some novel clues to understand the pathogenesis of HPND.
Recommend manuscript is edited by native English speaker.
Response: The manuscript has been edited by a native English speaker according to your suggestion.
Line 24 – This needs to be reworded, suggest Ultrastractural analysis of diseased crab tissues identified structural and …
Response: Ultrastractural analyses of diseased crab tissues were removed in the revised manuscript according to the suggestions of two reviewers in order to further highlight our research topic.
Line 41 – background on disease is required, what are the clinical signs?
Response: The clinical signs have been added in the revised manuscript.
Line 47 what is hydrops?
Response: The meaning of hydrops is ‘there is obvious effusion in the inner cavity of crab with HPND’. In order to reduce ambiguity, ‘hydrops’ has been removed in the modified version
Line 52 speciation? Can the authors clarify what they mean here.
Response: The meaning of speciation is ‘formation of species’. In the modified the sentence of ‘….., immunity and speciation due to presence……’ has been removed in the revised manuscript.
Line 63 Penaeus
Response: It has been modified.
Line 73 – what are the signs of infection? How was this confirmed to split the crabs between infected and uninfected? More details required
Response: The signs of sampled crabs have been described in detail according to your requirement.
Line 77 – More detail is needed, fixative used? Processing details? Stain? Suggest authors state samples were processed as reported by Yan et al 2020, if they describe full methods. Otherwise full details required here.
Response: The content for Electron microscopy observation has been removed in the revised manuscript according to the suggestions of two reviewers
Line 82 – were the same crabs sampled for TEM and sequencing?
Response: In this study, the crabs sampled from Yangcheng Lake, Aanfeng and Hainan towns were examined using a TEM. The sampled crabs from Hainan towns were used for metatranscriptomic sequencing. The crabs sampled for TEM and sequencing were same.
Line 152 – germplasm? Can the authors clarify what they mean here
Response: in the revised manuscript, ‘germplasm’ has been replaced with ‘offspring quality’.
Line 175 – Histology is usually wax sections observed under light microscope, study here describes TEM study suggest rewording unless histology was also used? In which case please detail results
Response: Thank you for your rigorous comments. The content for Electron microscopy observation has been removed in the revised manuscript according to the suggestions of two reviewers.
Line 176 onwards – The problem with studying tissues for presence/absence of a pathogen with TEM is the size of sample analysed, TEM takes very small pieces of tissue. We know some pathogens can be focal within tissues so could be missed when such small pieces of tissue are examined. This analysis can not be used to confirm that the tissues were pathogen free.
I also disagree with some of the conclusions and the images are not representative to display what the authors are suggesting, you ideally need lower power images to show damage, fixation of tissues looks poor and it is not clear whether it is the condition or fixation problems. Normal tissue structure needed as comparison.
Response: Thank you for your rigorous comments. The results of TEM observation has been removed in the modified version.
Tables– suggest labelling samples as healthy and diseased to assist the reader
Response: we have labeled samples as healthy and diseased following your suggestion
Figure 4 and 5 – suggest labelling as above
Response: we have labeled samples as healthy and diseased following your suggestion
Line 282 onwards – Interesting results, although the authors state viruses are not involved in HPND infection do the authors think the crabs are infected with these (or similar) virus infections? If so, could the presence of these viral sequences be related to possible stress and immune suppression? Are you more likely to see virus sequences in infected animals?
Response: That is an excellent question. However, in terms of the current data, it is difficult for us to draw a clear conclusion. Crab can be infected by Eriocheir sinensis reovirus found in this study according to previous study. We guess that some viruses, e.g Penaeus monodon nudivirus, Metopaulias depressus WSSV-like virus, infectious spleen and kidney necrosis virus, may infect crab.
Line 306 – sequencing did not record H. Eriocheir but could crabs be infected with another species of microsporidian? Figure 3B could be early development stages of a microsporidian infection.
Response: Yes. In this study, we did not found the H. eriocheir in the hepatopancreatic flora, but the sequences representing another species of microsporidian could be found in the hepatopancreatic flora, suggesting some microsporidian are pathogens of crab. Schizont-like of a microsporidian could be observed in the Figure 3B, but mitochondria can also found in the Schizont-like. It is known that Microsporidia has no mitochondria; therefore, we suggested that the structure observed in Figure 3B is not from microsporidium.
Line 385 – Do the authors think the crabs are infected with these (or similar) viruses? What is the significance of this finding?
Response: According to the principle of metatranscriptomic sequencing, the detected sequences are derived from transcripts of microbial genes. Therefore, we suggest that the detected DNA viruses may infect crab because these viral genes transcribe in the crab. However, whether these RNA viruses can infect river crabs needs further study. Our findings provide clues for molecular epidemiological investigations. It has been discussed in the modified version.
Line 407 – as above you cannot state this based on the limited TEM study performed
Response: we have modified description following your suggestion and removed the part of TEM as your suggestion.

Reviewer 3 Report
The work of Shen et al. deals with a new vision of the Hepatopancreatic Necrosis Disease of crabs, highlighting few aspect of disease pathogenesis. The bacterial pathogen Spiroplasma eriocheiris was also observed in diseased animal. The work is complex and well structured and the analysis performed interesting.
The metagenomic is complete and well performed, only few minor concerns should be elucidated:
- The authors show figures of macroscopic appearance of healthy and disease animals and the hepatopacreas. Figure 1 A-C are blurry. Please change them.
- Electron microscopy quality: TEM images are of poor quality and not acceptable in the present form.
Figure 2. the authors describe “cavitation of cytoplasm (CC), autophagosomes (AP), deformed mitochondria (MT) etc..” In my opinion samples were badly fixed and I can’t really see what they’re describing.
Figure 3: A. the image display the blade over cutting; C. the photo looks distorted.
I can see the cavitation of the cytoplasm in D but if the authors wanted to show the deformed mitochondria they need to put additional analysis. Better less images but more clear and of better quality.
Please consider to remake the 2 tables.
Author Response
Comments and Suggestions for Authors
The work of Shen et al. deals with a new vision of the Hepatopancreatic Necrosis Disease of crabs, highlighting few aspect of disease pathogenesis. The bacterial pathogen Spiroplasma eriocheiris was also observed in diseased animal. The work is complex and well structured and the analysis performed interesting.
The metagenomic is complete and well performed, only few minor concerns should be elucidated:
Response: Thank you for your positive comments; we have revised manuscript following your suggestion. We hope that all the changes will meet the requirements to make the manuscript acceptable for publication in Biology.
- The authors show figures of macroscopic appearance of healthy and disease animals and the hepatopacreas. Figure 1 A-C are blurry. Please change them.
Response: The qualities of figures 1A-C have been improved.
- Electron microscopy quality: TEM images are of poor quality and not acceptable in the present form.
Figure 2. the authors describe “cavitation of cytoplasm (CC), autophagosomes (AP), deformed mitochondria (MT) etc..” In my opinion samples were badly fixed and I can’t really see what they’re describing.
Response: The content of TEM observation have been removed based other two reviewer’s suggestions.
Figure 3: A. the image display the blade over cutting; C. the photo looks distorted.
I can see the cavitation of the cytoplasm in D but if the authors wanted to show the deformed mitochondria they need to put additional analysis. Better less images but more clear and of better quality.
Response: The content of TEM observation has been removed based other two reviewer’s suggestions.
Please consider to remake the 2 tables.
Response: Yes, we have done.
